# Discovery of Bactericidal Proteins from *Staphylococcus* Phage Stab21 Using a High-Throughput Screening Method

**DOI:** 10.3390/antibiotics12071213

**Published:** 2023-07-21

**Authors:** Ellisiv Nyhamar, Paige Webber, Olivia Liong, Özgenur Yilmaz, Maria Pajunen, Mikael Skurnik, Xing Wan

**Affiliations:** 1Department of Bacteriology and Immunology, Human Microbiome Research Program, Faculty of Medicine, University of Helsinki, 00290 Helsinki, Finland; 2Department of Microbiology, Faculty of Agriculture and Forestry, University of Helsinki, 00790 Helsinki, Finland; 3Faculty of Health Sciences, Kirklareli University, 39000 Kirklareli, Turkey

**Keywords:** bacteriophages, Hypothetical Proteins of Unknown Function (HPUFs), antimicrobial resistance, *Staphylococcus aureus*, cross-species toxicity

## Abstract

In the escalating battle against antimicrobial resistance, there is an urgent need to discover and investigate new antibiotic strategies. Bacteriophages are untapped reservoirs of such potential antimicrobials. This study focused on Hypothetical Proteins of Unknown Function (HPUFs) from a *Staphylococcus* phage Stab21. We examined its HPUFs for bactericidal activity against *E. coli* using a Next Generation Sequencing (NGS)-based approach. Among the 96 HPUFs examined, 5 demonstrated cross-species toxicity towards *E. coli*, suggesting the presence of shared molecular targets between *E. coli* and *S. aureus*. One toxic antibacterial HPUF (toxHPUF) was found to share homology with a homing endonuclease. The implications of these findings are profound, particularly given the potential broad applicability of these bactericidal agents. This study confirms the efficacy of NGS in streamlining the screening process of toxHPUFs, contributes significantly to the ongoing exploration of phage biology, and offers promises in the search for potent antimicrobial agents.

## 1. Introduction

Antimicrobial resistance is one of the most pressing challenges facing modern medicine. However, the rate of discovery and introduction of new classes of antibiotics has slowed drastically over the decades, with only two introduced to the market since 1962 [1]. The need for new antibiotic classes is only increasing as the potential for analogue development from existing classes is depleted. Pathogens that exhibit alarming resistance against current antimicrobial treatments are given the acronym of ESKAPE, including *Enterococcus faecium*, *Staphylococcus aureus*, *Klebsiella pneumoniae*, *Acinetobacter baumannii*, *Pseudomonas aeruginosa*, and *Enterobacter* species [2]. Among them, methicillin and vancomycin-resistant *Staphylococcus aureus* (MRSA/VRSA) are placed in the high-priority category by the World Health Organization to accelerate the development of new antimicrobial compounds for treating multi-drug-resistant *S. aureus* [3]. MRSA is usually acquired within a hospital environment and can cause various severe opportunistic infections and diseases [4]. Vancomycin has long been regarded as a drug of last resort for MRSA treatment. Unfortunately, the emerging VRSA jeopardises the efficiency of clinical treatments and leads to a significant increase in staphylococcal bacteraemia mortality [5].

Bacteriophages, especially lytic phages, are viruses that infect, propagate, and destroy bacteria. Throughout evolution, phages continuously adapt to confront their bacterial hosts for their own survival. This intertwined relationship endows the phages with highly specific mechanisms to reprogram bacterial cell metabolism. Phage-inspired antibacterial target discovery is another ascending approach to harnessing the antimicrobial activity of phages [6]. Identifying novel bacterial targets through phage research could help screen and design new small molecular compounds that replicate the growth-inhibitory effects of the antibacterial phage proteins. Intriguingly, a substantial proportion of phage gene products have completely unknown functions since they have not been characterized and have sequences that do not correspond with any proteins of known function. Screening hypothetical proteins of unknown function (HPUFs) from bacteriophages for toxic activity against bacteria may provide new and potentially life-saving approaches to combat bacterial infections [7,8]. An earlier extensive study on mining 26 *S. aureus* phage genomes revealed that 31 polypeptide families showed toxicity towards the bacterial host, including ORF104 from phage 77 [5]. The gene product of ORF104 interacted with bacterial essential protein DnaI, disrupting DNA synthesis and cell growth. The research team then used the protein pair to screen for small molecular inhibitors, and identified 36 of them interrupting the ORF104-DnaI interaction and 11 that were directly toxic against *S. aureus* [5].

Stab21 is a lytic *Staphylococcus* phage recently discovered by Oduor et al. [9]. The phage is taxonomically classified to the *Kayvirus* genus of the Twortvirinae subfamily in the Herelleviridae family (accession number LR215719, [9]). Stab21 has a wide host range and possesses no genes associated with antibiotic resistance or virulence [9]; therefore, it is a good candidate for therapeutical applications. Of all 238 predicted genes in the Stab21 genome, 203 were preliminarily annotated to encode HPUFs. Many lytic phages could possess genes encoding proteins that directly or indirectly mediate the destruction of their host bacterial cells [10]. Hence, among the HPUFs of Stab21, there is a great potential for the discovery of bactericidal proteins that alter bacterial pathways in an unprecedented manner.

The use of a next-generation sequencing (NGS)-based screening approach to screen for phage-encoded toxic proteins is shown to be as reliable as the alternative plating-based toxicity screening method. An earlier direct comparison was performed during the development of the NGS method, where it was concluded that the NGS-based assay not only provides similar screening results as the plating-based assays, but is also superior in efficiency, accuracy, and reliability [11,12].

In the study presented here, we investigated all hypothetical proteins and selected 96 true HPUFs for their bactericidal activity in *E. coli* using the NGS-based approach. We confirmed their bactericidal activities in *E. coli* and their potential anti-*S. aureus* abilities using various inducible expression systems. Five of the Stab21 gene products were shown to have cross-species toxicity towards *E. coli*. Through detailed in silico functional and structural analyses, we found that Gp024 shares high structural homology with a homing endonuclease. Our work not only underscores the potential of Stab21 HPUFs as antimicrobial agents, but also reinforces the efficacy of NGS in identifying these potential assets. 

## 2. Results

The preliminary annotation of the Stab21 genome, based mainly on the Basic Local Alignment Search Tool (BlastP), identified 203 HPUFs [9]. This number was refined to 96 HPUFs after a more detailed examination through HHPred [13] searches and the identification of phage-particle-associated proteins by liquid chromatography–tandem mass spectrometry (LC-MS/MS) analysis [14]. 

### 2.1. Potential Toxicity of Stab21 Gene Products

The bactericidal potential of the 96 selected HPUF genes of Stab21 was investigated using the NGS approach [11]. The sum of four ligation joint sequences for each gene was calculated and used as its total read coverage. The relative number of joint sequence reads was calculated for all 96 genes by dividing the total read coverage of a single gene by the total number of joint sequence reads for all genes in the pool and expressed as a percentage (Appendix A). 

This method identified 16 gene products with toxic potential characterized by ratios under 0.5. Genes *g062c*, *g085*, and *g081c* exhibited the three lowest ratios. Additional 31 HPUF-encoded gene products with ratios between 1.0 and 0.5 were identified as potentially mildly toxic (Figure 1), while the remaining 42 genes with ratios above 1.0 were deemed non-toxic (Appendix A).

Genes *g002* and *g018*, which had no detected ligation joint reads in the plasmid mixture, could not be conclusively evaluated. It is possible that the gene fragments were not successfully digested by restriction enzymes, leading to the absence of a suitable restriction site for ligation to the vector. 

Ratios of relative reads for all the analysed HPUFs ranged between 0.0002 (for *g062c*) and 8.5133 (for *g196*). The complete results, including exact ratios and relative vector gene joint reads in both ligation mixtures and transformant plasmids, are listed in Appendix A.

### 2.2. Five Stab21 Genes Products Inhibit the Growth of E. coli

A subsequent screening of the effect of toxHPUF candidates on *E. coli* DH5α was conducted via a drop test after cloning the candidate genes into the expression vector pBAD33. Arabinose-induced expression of the Stab21 HPUFs resulted in significant *E. coli* growth inhibition by 9 of the initially identified 16 genes, although with varying toxicity levels (Figure 2). Particularly, the gene products Gp081c, Gp159, Gp175, Gp209, Gp212, and Gp213 strongly suppressed the *E. coli* growth. Additionally, Gp172 seemed to diminish the overall viability of *E. coli*, as fewer colonies formed even in the presence of glucose compared to other nontoxic gene products. Gene products Gp024 and Gp187, on the other hand, only demonstrated mild toxicity to the host.

Employing a more stringent induction expression system with an anhydrotetracycline-inducible promoter [15], we followed the growth curves of the *E. coli* strains carrying these recombinant genes on the pRAB11N plasmids. 

The drop test indicated that nearly all nine candidates displayed varying toxicity levels under inducing conditions of anhydrotetracycline (ATc) 0.4 µM (Appendix A). However, growth curve analysis offered a more reliable interpretation, ruling out false positives caused by mutagenesis from overnight incubation. As shown in Figure 3, after inducing the Stab21 HPUF genes with ATc, five genes significantly impeded the growth of *E. coli*, specifically genes *g024*, *g081c*, *g172*, *g187* and *g213*. Yet, after six hours, resistant mutants began to appear and resume the growth. 

### 2.3. Stab21 Gp081c Might Also Inhibit the Growth of S. aureus

The toxicity of Stab21 HPUFs on *S. aureus* was investigated via a drop test. Initially, chloramphenicol (Cm) was used to preserve the pRAB11N-HPUF plasmids in *S. aureus*. However, using both Cm and ATc seemed to be toxic to the host *S. aureus* strains, as the combination of Cm30 (30 µg/mL chloramphenicol) and ATc 0.4 µM inhibited even the growth of RN4220/pRAB11N. Similar inhibitory effect was also observed when using lower Cm concentration (Appendix A). Decreasing the ATc concentration, on the other hand, did not seem to be sufficient for the HPUF expression (Appendix A). Therefore, we decided to leave Cm out, and used only the ATc inducer. Under these conditions, upon induction, Gp024, Gp081c, Gp175, Gp187, Gp209, Gp212 and Gp213 showed toxicity (Figure 4).

We also obtained growth curves for all strains under both ATc-induced and non-induced conditions in liquid cultures. In *S. aureus* RN4220, no HPUFs had a significant toxic effect when induced, but the known toxic gene product of ORF104 [7] did exhibit toxicity, even under the non-induced condition (Figure 5a). Gene products of *g081c* slightly delayed the *S. aureus* growth; however, the growth pace swiftly recovered due to the emergence of rescue mutants (Figure 5b).

### 2.4. Non-Inducing Mutation of g172 and g187 Tolerant Clones Located in tetR of pRAB11N

Upon induction, five gene products, Gp024, Gp081c, Gp172, Gp187, and Gp213, showed toxicity towards *E. coli*. However, after 6 h, the growth of toxin-tolerant mutants was observed from all recombinant strains. We suspected that the tolerant phenotype may have resulted from mutations in the plasmid DNA by inactivating the toxin, or in the *E. coli* genomic DNA, potentially providing a clue of the identity of the target for the toxin. Therefore, to study this, we cultivated *E. coli* DH10B/pRAB11N-toxHPUFs in the presence of ATc for 16 h to allow the emergence of toxin-tolerant mutants. The presence of plasmids was examined by polymerase chain reaction (PCR). The plasmid was only detected in tolerant clones (denoted with ‘-T’ suffix) recovered from DH10B/pRAB11N-*g172*-T and DH10B/pRAB11N-*g187*-T (Appendix A). Subsequent isolation and sequencing over the HPUF gene insertion of the plasmids pRAB11N-*g172*-T and pRAB11N-*g187*-T revealed that the inserts were 100% identical to the original toxic HPUF gene sequences. This indicated that no mutations in the toxHPUF genes could explain the tolerant phenotype. 

Given this result, we re-evaluated the toxicity of Gp172 and Gp187 in the toxin-sensitive strains DH10B/pRAB11N-*g172* and DH10B/pRAB11N-*g187* and in the toxin-tolerant mutant strains DH10B/pRAB11N-*g172*-T and DH10B/pRAB11N-*g187*-T. Growth curves confirmed that upon ATc-induction the toxin-sensitive strains were inhibited up to 6 hr, and that the toxin-tolerant mutants remained non-toxic (Appendix A).

To identify the genomic mutation behind the toxin tolerance, the genomes of DH10B/pRAB11N-*g172*, -*g187*, -*g172*-T, and -*g187*-T were sequenced and subjected to de novo assembly. The obtained contigs were then compared to both the host genomic and the plasmid sequences. While the genomic sequence contigs of the four *E. coli* strains were 100% identical to the *E. coli* DH10B reference sequence (GenBank accession no. NC_010473), the plasmid sequences of DH10B/pRAB11N-*g172*-T and -*g187*-T carried both a single nucleotide mutation when aligned with the plasmid pRAB11N-*g172* and -*g187* sequences. Specifically, an A to C mutation was present at position 3654 in pRAB11N-g*172*-T, and a G to T mutation at position 3747 in pRAB11N-*g187*-T. Both mutations map into the *tetR* gene, causing Leu-113-Arg and Asp-82-Lys substitutions, respectively (Appendix A). 

### 2.5. Gp024 Is Homologous to a Homing Endonuclease

The Phyre2 software (Version 2.0, http://www.sbg.bio.ic.ac.uk/~phyre2, accessed on 26 June 2023) was used to predict the functions of toxHPUFs Gp024, Gp081c, Gp172, Gp187, and Gp213. In Phyre2, a reliable model typically has an input and template sequence identity of over 30% and a confidence level of more than 90% [16]. Eighty residues (37% sequence coverage) of Gp024 were modelled with a 98.0% confidence to the C-terminus of *Bacillus* phage SPO1 homing endonuclease I-HmuI chain M (Protein Data Bank (PDB): c1u3eM). For *g081c*, a confidence of 85.5% was observed for 53 residues (a 48% sequence coverage), which matched to the *Saccharomyces cerevisiae* monopolin complex protein subunit CSM1. However, Gp172, Gp187, and Gp213 yielded models with low confidence scores, below 51%.

The HHpred [13] software was also used to predict functions of the same toxHPUFs by searching each toxHPUF against PDB mmcif70. In the HHpred analysis, a higher score value signifies a better sequence alignment quality and similarity. An expected value (E-value) close to zero indicates a strong identity by estimating the number of similar score hits that could occur randomly. Probability values further underscore the statistical significance of sequence and hit homology [17]. The best hit for Gp024 was also *Bacillus* phage SPO1 homing endonuclease I-HmuI (PDB: 1U3E) with a probability of 98.18%, a score of 42.4 and an *E*-value of 0.023. The Gp081c top model was *Enterobacter* phage 22 tail needle protein Gp26 with a probability of 95.31%, a score of 37.3 and an *E*-value of 0.21. The top model of Gp187 was *Homo sapiens* zinc finger protein SNA1 with a probability of 99.11%, a score of 75.52 and an *E*-value of 9.2^−11^. The homologous models of Gp172 and Gp213 resulted in low probability and *E*-values.

Finally, we employed AlphaFold2 [18] to predict the structure of the five toxHPUFs (Appendix A). PyMOL was used to superimpose the functional models from Phyre2 and HHpred with the structural models of AlphaFold2. The root mean square deviation (RMSD) score measures the structural overlap between the two input models using the distance between aligned α-carbon atoms, with a score below 2.5 Å considered as an acceptable alignment [19]. When superimposing the truncated *Bacillus* phage SPO1 homing endonuclease I-HmuI (c1u3eM) obtained from the best Phyre2 model with the Gp024 AlphaFold2 model, the RMSD score was 1.581 Å between 138 atoms of the two proteins. Two C-terminal α-helixes from both proteins were found overlapping with each other (Figure 6a). When superimposing the best homologous model of I-HmuI (1U3E) from HHpred analyses with the predicted structure of Gp024, the RMSD score was 2.217 Å when aligning 181 atoms. The N-terminal DNA-binding surface of I-HmuI loosely overlapped with the N-terminus of Gp024 by two β-sheets and one α-helix (Figure 6b). Alignment of Gp081c with both Phyre2 and HHpred functional models resulted in RMSD scores of over 8 Å (Appendix A). The predicted structure obtained from AlphaFold2 for Gp187 was composed only of two β-sheets and so was omitted from further analysis.

## 3. Discussion

The escalating prevalence of antibiotic resistance in critical pathogens such as *S. aureus* demands the discovery of novel antimicrobial molecules [20]. Phages, with their unique genetic diversity, present a promising reservoir of potential antimicrobials, yet the majority of their genomes remain uncharacterized. Current phage research identifies a range of bacteriotoxic molecules such as endolysins and polysaccharide depolymerases that could potentially be harnessed against antibiotic resistant bacterial infections [21,22,23]. Given the wide genomic variation even among phages infecting the same host, there is significant potential for the discovery of new antimicrobial mechanisms and targets from the multitude of gene products currently deemed HPUF [24]. Such discoveries may yield innovative antibiotic treatments capable of countering the ever-increasing threat of antimicrobial resistance [6].

In this study, we employed an NGS-based screening assay to identify toxic proteins encoded by *Staphylococcal* phage Stab21. The HPUFs of Stab21 were screened for bacteriotoxic effects against *E. coli* and their bactericidal activities were confirmed in both *E. coli* and *S. aureus*. Our findings revealed that five gene products were toxic to *E. coli* DH10B using arabinose- and anhydrotetracycline-induced expression vectors. Crucially, this study verified that the application of this high-throughput NGS-based screening process in *E. coli* is not limited to examining HPUFs from only phages infecting Gram-negative bacteria. 

Our study employed an already established protocol for NGS-based screening in *E. coli* [11,12], recognizing the value of this model organism in providing insights into cellular mechanisms that transcend species boundaries [25]. By adopting this protocol, we anticipate a rapid determination of any bacteriotoxic activity of HPUFs in both Gram-negative and Gram-positive strains.

In this context, we constructed the pCU1LK vector which contains the backbone of the *E. coli*–*S. aureus* shuttle vector pCU1 [26] that can accommodate up to 6 kb fragments cloned under the *lac* promoter. As the size of the inserted linker sequence was only 45 bp in pCU1LK, it should also allow cloning up to 6 kb fragments without problems to replicate freely in both hosts. 

However, as the *lac* promoter is notoriously known as leaky, we selected another tighter expression vector pRAB11 with two *tet* operator *tetO* sites [27] to verify the toxicity of HPUFs. Nevertheless, we observed an unexpected growth inhibition of *S. aureus* strain RN4220/pRAB11N in the presence of ATc on solid media (Appendix A). This effect was not noticed in *E. coli* DH10B/pRAB11N. We do not have a clear explanation for this phenomenon. Despite the expectation that ATc should induce the gene expression at lower concentrations and with less toxicity than its parent antibiotic, tetracycline (Tc) [28], aged ATc has been found to generate toxic breakdown products that can interfere with *S. aureus* growth [29]. It is possible that these breakdown products are less effective on *E. coli* than on *S. aureus*, as observed in our study.

Though our study did not identify any toxHPUFs towards *S. aureus* among the nine *E. coli* -toxic candidates, further examination should be carried out for the remaining 87 HPUFs of Stab21, which were regarded as non-toxic to *E. coli*. That could involve pooling pCU1LK-HPUFs or pRAB11N-HPUFs ligation mixtures and subsequently screening successful transformants for toxicity-presenting an alternative to the NGS-based screening method. Alternatively, an arsenite-inducible plasmid, pT0021, which has been previously employed effectively to screen for toxHPUFs in *S. aureus* [7], could serve as another viable option for this investigation. 

Furthermore, for future screening of HPUFs from various phages derived from different bacteria, the NGS screening technique presented in our study can be fine-tuned depending on the bacterial host by applying different inducible plasmid vectors such as the Tn7-based integration vector pTNS2 [30] for *Pseudomonas aeruginosa*, or vectors derived from the IncQ plasmid for *Acinetobacter baumannii* [31]. 

Among the structural studies of the top five identified toxic HPUFs, only Gp024 yielded a reliable protein model, aligning with the *Bacillus* phage SPO1 homing endonuclease I-HmuI. Although the predicted structure of Gp024 resembles different domains in HHpred and Phyre2, its function as a homing endonuclease cannot be completely ignored. Previous studies have identified phage HPUFs as homologs to homing endonucleases. A noteworthy example is found in the extensively researched *Escherichia* phage T4, which houses 15 homing endonuclease encoding genes, showing the evolutionary link between these proteins and phage biology [32]. Adding to the intrigue is the existence of colicins, DNases produced by *E. coli* that demonstrate toxicity against other *E. coli* strains. These colicins belong to the same H-N-H family of endonucleases [33] as the suspected homing endonuclease Gp024 of Stab21. The analogous functions and structural similarity of these proteins could suggest that Gp024 could exhibit characteristics of homing endonucleases like colicins. However, further experiments such as single-site mutations and DNA nicking assays are needed to confirm its role as a homing endonuclease.

Our findings that five HPUFs from Stab21 exhibit cross-species toxicity towards *E. coli* have profound implications. It is possible that these HPUFs share a conserved molecular target in both *E. coli* and *S. aureus*, significantly broadening the potential applicability of these bactericidal agents. In conclusion, our research opens a promising avenue for the discovery of novel, potent antimicrobial agents, providing hope in the fight against growing antimicrobial resistance.

## 4. Materials and Methods

### 4.1. Bacterial Strains, Plasmids, Phage and Culture Conditions

All bacterial strains and plasmids used for experimentation are listed in Table 1 and Appendix A. Commercial electrocompetent *Escherichia coli* DH10B cells (Thermo Fischer Scientific, Waltham, MA, USA), in-house prepared electrocompetent *E. coli* DH5ɑ and in-house prepared electrocompetent *Staphylococcus aureus* RN4220 were used as expression hosts. 

*E. coli* DH5ɑ, DH10B and their derivatives were grown in Lysogeny broth (LB; 10 g/L Tryptone (Neogen, Lansing, MI, USA, Cat no. NCM02118A), 5 g/L Yeast Extract (Neogen, Cat no. NCM0218A), 10 g/L NaCl) or on agar (LA, LB supplemented with 1.5% Bacto agar). For toxicity tests, *E. coli* strains were grown in M9 minimal media (KH_2_PO_4_ 3 g/L, 0.5 g/L NaCl, 6.78 g/L Na_2_HPO_4_, 1 g/L NH_4_Cl, casamino acid 0.2% (*v*/*v*), MgSO_4_ 2 mM, CaCl_2_ 0.1 mM, thiamine 1 mg/L) supplemented with antibiotic to maintain the plasmid, and glucose for repression and arabinose for induction. *S. aureus* strains were grown in Tryptic Soy Broth (TSB; VWR Chemicals, Radnor, PA, USA, Cat. No. 470015-844) or on Tryptic Soy Agar (TSA, Vegitone, Sigma-Aldrich, St. Louis, MA, USA, Cat. No. 14432) or broth (Dehydrated TSB, VWR Chemicals). Liquid cultures were grown at 37 °C overnight with 200 RPM shaking unless stated otherwise. Solid cultures were incubated at 37 °C overnight. To maintain the plasmids, broth or agar was supplemented with either 100 µg/mL of ampicillin (Amp100) or 30 µg/mL of chloramphenicol (Cm30) unless stated otherwise. 

### 4.2. DNA Manipulations

For plasmid isolations, individual colonies were obtained on a streak plate, and 1 colony was used for inoculation to obtain overnight cultures. Plasmids from *E. coli* strains were extracted, purified, and precipitated with either NucleoBond™ Xtra Midi kit and NucleoBond™ Finalizers (Machery-Nagel, Düren, Germany) from 200 mL overnight cultures or with the NuceloSpin Plasmid EasyPure Kit (Machery-Nagel) for 1 mL cultures, according to manufacturers’ instructions. To isolate the plasmids from Gram-positive *S. aureus*, lysostaphin lyophilized powder from *Staphylococcus staphylolyticus* (Merck KGaA, Darmstadt, Germany) was added to the final concentration of 20 µg/mL and incubated for 1 h at 37 °C before using NucleoSpin Plasmid EasyPure Kit according to the manufacturer’s manual. For the Illumina sequencing, plasmid pools were extracted with NucleoBond™ Xtra Midi kit, and eluted in a 200 µL Tris/HCl pH 8.5 elution buffer. 

*E. coli* and *S. aureus* genomic DNA was isolated using the JetFlex Genomic DNA Purification Kit (Thermo Fischer Scientific) following the bacterial gDNA isolation protocol for Gram-negative and Gram-positive bacteria, the latter using a 20 µg/mL lysostaphin pre-treatment. The isolated DNA was rehydrated at 22 °C for 16 h. 

DNA fragments were amplified by PCR using primers listed in Appendix A. Stab21 phage DNA [14] was used as a template for amplification of HPUFs. *S. aureus* Newman gDNA was used to amplify the toxic control gene ORF104. Plasmid DNA and colonies were used as templates for confirmation of correct gene insertion. Phusion High-Fidelity DNA Polymerase (Thermo Fischer Scientific) was used for generating DNA fragments with the highest accuracy for cloning, while DreamTaq DNA Polymerase (Thermo Fisher Scientific) was used for screening the presence of a certain DNA fragment either from the colony or from the ligation mixture. The PCRs were run in a T100™ or iCycler Thermal Cycler (Bio-Rad Laboratories, Inc., Hercules, CA, USA) following standard manufacturer protocol for the polymerases. 

All restriction enzymes used in this study were obtained from Thermo Fisher Scientific or New England Biolabs (MA, USA). Plasmid vectors were linearised with restriction enzymes as stated according to the manufacturer’s instructions, and dephosphorylated with FastAP™ Thermosensitive Alkaline Phosphatase (Thermo Fisher Scientific) at 37 °C for 30 min followed by a 15 min heat inactivation at 65 °C.

Sticky-end ligation of double-digested individual HPUF-encoding gene fragments to linearised and dephosphorylated pCU1LK or pRAB11N vectors was carried out at a 1:3 vector to insert molar ratio, and the total DNA concentration was adjusted to 10 ng/µL. T4 DNA Ligase (5 U) (Thermo Fischer Scientific) was used for all the ligation reactions. The ligation reaction was incubated at room temperature overnight (15 h) before heat inactivation at 65 °C for 10 min. 

NucleoSpin Gel and PCR Clean-up XS kit (Machery-Nagel) was used to purify and concentrate DNA after PCRs and enzymatic reactions.

For the NGS screening assay, the Stab21 HPUF-encoding genes and pCU1LK vector were first double digested with restriction enzymes NotI and NheI or KpnI FastDigest™ enzymes (Thermo Fisher Scientific) depending on the insertion fragments (Appendix A) [11]. Every 16 ligation mixtures of the HPUF gene and vector pCU1LK were pooled before concentration by kit, and an elution volume of 20 µL in Baxter Sterile Water (Baxter Corporation, Deerfield, IL, USA) was used per pool. One microliter (ca. 200 ng) of each ligation pool was transferred to 50 µL of *E. coli* DH10B cells through electroporation. Plasmids from transformation reactions were isolated from a 3 h culture inoculated with all colonies formed on the transformation plates. DNA samples from both the ligation pool and the plasmid pool were sequenced with the 150 bp paired-end protocol in the Illumina HiSeq platform at NovoGene Company Ltd. (Cambridge, UK) as described by Kasurinen et al. [11].

### 4.3. Electroporation and Transformation

Electroporation was performed with a Gene Pulser™ apparatus (Bio-Rad Laboratories) using 0.2 mm cuvettes. For the transformation of *E. coli* strains, the parameters of 200 Ω resistance, 25 mF capacitance and 2.5 kV voltage resulted in a time constant between 4.5 and 5.0 ms. Transformed *E. coli* cells were recovered in a 1 mL super optimal broth (SOC; 2% Tryptone, 0.5% Yeast Extract, 10 mM NaCl, 2.5 mM KCI, 10 mM MgCl_2_, 10 mM MgSO_4_, 20 mM glucose) and incubated at 37 °C with a 200 rpm shaking for 45 min before being plated on LB Amp100 agar plates using 10 µL, 100 µL, and the remainder of cells collected through centrifugation. For NGS screening, every 50 µL of the recovered cells from each pool were spread onto LB Amp100 plates, resulting in 20 plates. The plates were incubated at 37 °C overnight. 

For the transformation of *S. aureus* RN4220, the parameters of the Gene Pulser electroporator were set with a resistance of 100 Ω, a capacitance of 25 µF, and a voltage of 2.3 kV resulting in a time constant between 2.0 and 2.4 ms. Transformed *S. aureus* were recovered in a 850 µL TSB and a 150 µL 2 M sucrose and incubated at 37 °C with a 200 rpm shaking for 90 min. The same plating scheme and incubation conditions were used as those for *E. coli* electroporation. 

### 4.4. Construction of Vectors pCU1LK and pRAB11N

To construct a vector suitable for screening the toxicity of all the 96 HPUFs in *E. coli*, we added a linker to the pCU1 plasmid. The KpnI-PstI linker containing restriction sites for BamHI, XbaI, NheI, NcoI, and NotI was constructed by annealing oligonucleotides NOTup and NOTdown (Appendix A) with a final concentration of 50 μM in a 20 μL linker solution (50 mM Tris-HCl pH 8.0, 100 mM NaCl, 1 mM EDTA). The reaction mixture was incubated at 95 °C for 2 min, followed by 10 min at 52 °C. The annealed linker was phosphorylated with T4 polynucleotide kinase (T4 PNK, Thermo Fisher Scientific) before ligation to gel-purified pCU1 linearised with KpnI and PstI digestions. The resultant vector pCU1LK was used for the preliminary screening of the toxicity of all HPUFs in *E. coli*.

The *E. coli–S. aureus* shuttle vector pRAB11N was re-constructed to make an exact copy of the plasmid pRAB11, which could no longer be obtained from any source. The plasmid pRMC2 [34] was used as a template in plasmid-PCR using pRAB-fw and pRAB-rev as primers (Appendix A). This PCR added a second *tet* operator to the *tetR* promoter region. A 30 µL aliquot of the obtained PCR product was digested with a 50 U DpnI (New England BioLabs, cat. no. R0176) in a reaction volume of 50 µL to eliminate the pRMC2 template before ligation. One of the confirmed transformants was named pRAB11N, and it was used as a shuttle expression vector in both *E. coli* and *S. aureus*. 

The correctness of the obtained plasmid vectors was confirmed by Sanger sequencing at the Finnish Institute for Molecular Medicine (FIMM) Genomics Sequencing (Biomedicum, Tukholmankatu 8, Helsinki, Finland) using primers Puc19-F and Puc19-R for pCU1LK, and fR-346 for pRAB11N. Primers used in this study are listed in Appendix A.

### 4.5. Bioinformatics

The NGS-based toxicity screening of HPUF encoding genes was carried out using the protocol described previously [11]. The DNA samples used for NGS are described in Section 2.3. For each pool of 16 HPUFs, the reads containing the four expected ligation joint sequences (VGF, GVF, VGR, and GVR, Figure 7, and Appendix A) were identified and extracted from both the pooled ligation mixture DNA and the pooled plasmid DNA samples using the script and workflow described earlier [11]. The total number of the four ligation joint sequences for each HPUF-encoding gene was calculated and used to represent their total read coverage (N joint reads, Formula (1)). The relative number of joint sequence reads was calculated for all genes in the pools by dividing the total read coverage of a single gene by the total number of joint sequence reads for all genes in the pool and expressed as a percentage (relative gene percentage, Formula (1)). As described by Kasurinen et al., a low ratio (Formula (2)) between the relative joint sequence reads of individual genes from plasmid pools and those from the corresponding ligation mixture indicates the presence of toxHPUF gene, owing to the elimination of transformants carrying a toxic gene [11].
(1)$$Relative\;gene\;percentage=\frac{N\;joint\text{-}reads\;of\;single\;gene}{N\;joint\text{-}reads\;of\;all\;genes\;in\;pool}\times 100\%,$$
(2)$$Ratio=\frac{Relative\;gene\;percentage\;from\;pooled\;plasmid\;DNA}{Relative\;gene\;percentage\;from\;ligation\;mixtures}.$$

On the contrary, a ratio close to 1 or above indicates a non-toxic gene, reflecting the successful replication of the recombinant plasmid. In theory, a HPUF could be considered as bactericidal if this ratio is less than 1. In our study presented here, gene products exhibiting ratios between 0.5 and 1.0 were considered potentially mildly toxic, while those with a ratio under 0.5 were potentially bactericidal.

### 4.6. Toxicity Confirmation

The toxicity of the potentially toxic HPUFs towards host bacteria was primarily tested using a drop test on agar plates. Potentially toxic HPUFs genes were cloned either into the pBAD33 vector under an arabinose-inducible promoter (using the KpnI-XbaI restriction sites) or pRAB11N under an ATc-inducible promoter (using the BglII-KpnI sites). The correct gene insertions were confirmed by colony PCR using primers pBADF and pBADRev for pBAD33 constructions and pRAB11-F and fR-346 for pRAB11N constructions. The resultant PCR products were further analysed and identified by Sanger sequencing.

Subsequently, *E. coli* or *S. aureus* transformants carrying the desired recombinant plasmids (Appendix A) were grown on suitable agar plates to obtain single colonies. Three isolated colonies containing the recombinant vector and positive-control colonies were resuspended and diluted to optical density at 600 nm (OD_600_) 0.2 with sterile phosphate-buffered saline (PBS) pH 7.4. Five microliters of the serial dilutions from 10^−1^ to 10^−8^ of each culture were spotted onto both induced and non-induced LA or TSA plates and let dry before incubation at 37 °C overnight. Varying inducing and non-inducing conditions of the plates are listed in Appendix A.

To confirm toxicity of the HPUFs and visualise the time-specific impact of toxHPUFs, the growth curves of the strains carrying the pRAB11N-HPUF plasmids were determined with and without induction. Three colonies of each strain were inoculated in a 1 mL TSB supplemented with Amp100 for *E. coli*, and TSB supplemented with Cm30 for *S. aureus* and incubated at 37 °C overnight with a 200 RPM shaking. Cells were resuspended and washed with an equal volume of TSB, and OD_600_ was measured. The washed cells were used as a starting inoculum after resuspending to an OD_600_ of 0.01 into fresh TSB, with and without the inducer (ATc 0.4 µM). The bacteria were grown in Bioscreen Honeycomb plates (Oy Growth Curves Ab Ltd., Helsinki, Finland) in triplicate. The OD_600_ was measured using the Bioscreen C MBR (Oy Growth Curves Ab Ltd., Helsinki, Finland) every 30 min for 16 h with settings of continuous shaking, high amplitude, and normal speed. The shaking was stopped 10 s before each OD_600_ measurement. The mean values and standard deviations were calculated using triplicate data points.

### 4.7. Genomic DNA Sequencing

Bacterial gDNA isolated from *E. coli* DH10B/pRAB11N-HPUF clones was sequenced at Novogene Company Ltd. Contigs were assembled using BV-BRC (https://www.bv-brc.org/app/Assembly2, accessed on 26 June 2023), and a nucleotide BLAST (Basic Local Alignment Search Tool, version 2.14.0, https://blast.ncbi.nlm.nih.gov/Blast.cgi, accessed on 26 June 2023) search was used to align the sequence assemblies with the individual pRAB11N-HPUF and *E. coli* DH10B genome sequences. Protein BLAST was used to align sequences to known proteins.

### 4.8. Structural and Functional Analysis of Toxic Proteins

Phyre2, HHpred, and AlphaFold2 were used to predict the functions and structures of the toxic hypothetical proteins. Comparisons were made between the sequence identities of the proteins and known protein structures, with cut-offs of 30% identity and confidence levels over 90% for Phyre2 and similar parameters for HHpred. Functional and structural protein database files were superimposed using the molecular visualisation system PyMOL utilising the ‘super’ function (The PyMOL Molecular Graphics System, version 2.5, Schrödinger, LCC). RMSD scores were calculated using PyMOL to measure structural alignment and overlap of the predicted protein structures.

## Figures and Tables

**Figure 1 antibiotics-12-01213-f001:**
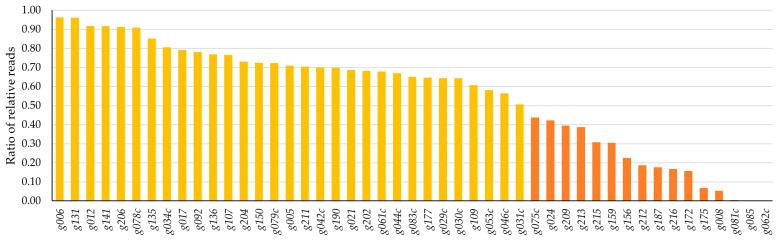
Results from the toxicity analysis using an NGS-approach of Stab21 phage HPUFs featuring ordered ratios of relative joint sequence reads for all 16 potentially toxic (orange bars) and 31 potentially mildly toxic genes (yellow bars). Non-toxic genes are elaborated in Appendix A.

**Figure 2 antibiotics-12-01213-f002:**
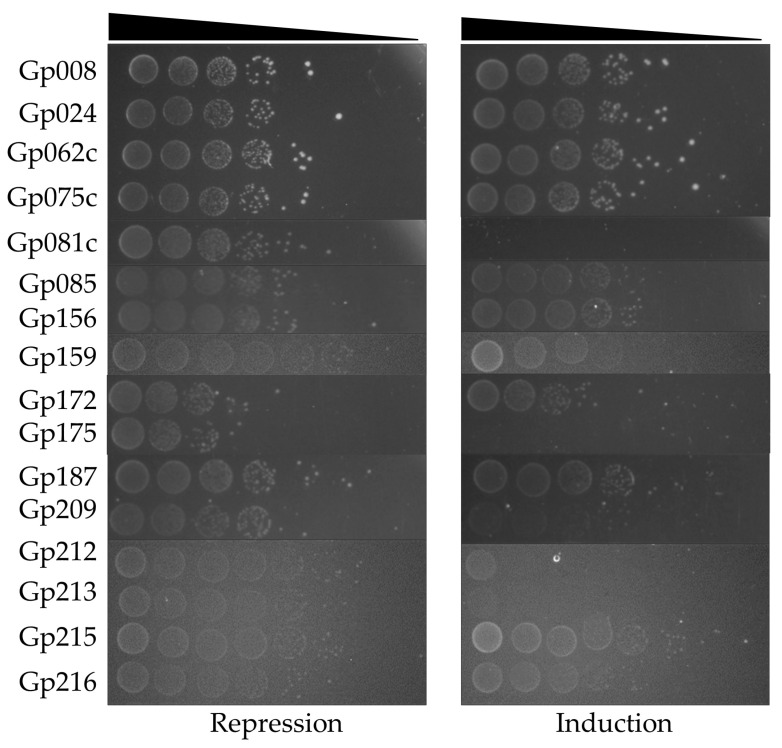
Initial drop test results of *E. coli* DH5α/pBAD33-HPUFs under repression (0.2% glucose) and induction (2% arabinose) conditions. Cell density from 10^−1^ to 10^−8^ dilutions is indicated with wedge.

**Figure 3 antibiotics-12-01213-f003:**
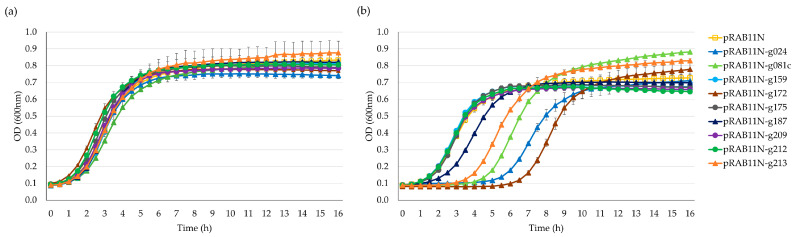
Growth curves of *E. coli* DH10B/pRAB11N-HPUFs for toxicity screening. (**a**) Toxicity screening without anhydrotetracycline (ATc) induction. (**b**) Toxicity screening with ATc 0.4 µM induction. Error bars show ± standard deviation calculated using triplicate results. OD, optical density.

**Figure 4 antibiotics-12-01213-f004:**
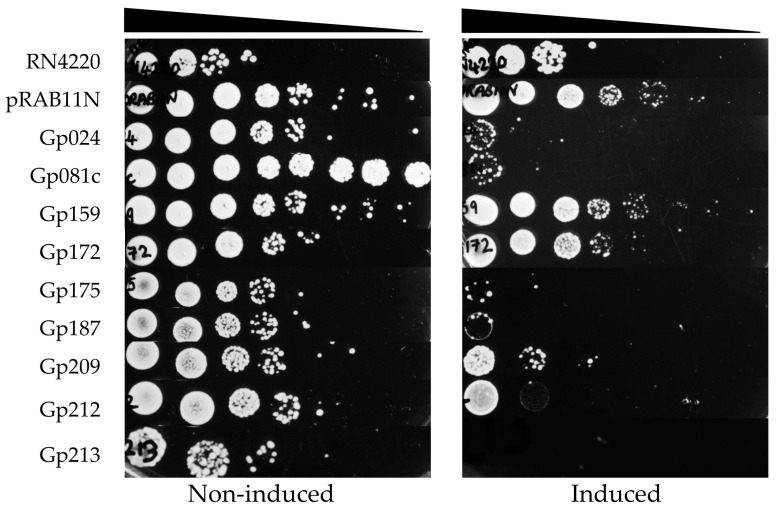
*S. aureus* RN4220/pRAB11N-HPUFs toxicity drop test under non-induced and induced conditions (ATc 0.4 µM). ATc, anhydrotetracycline. Cell density from 10^−1^ to 10^−8^ dilutions is indicated with wedge.

**Figure 5 antibiotics-12-01213-f005:**
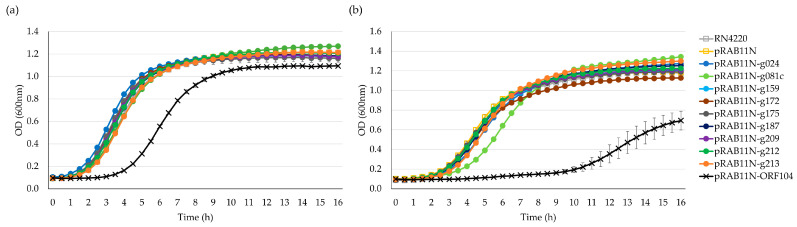
Growth curves of *S. aureus* RN4220 pRAB11N-HPUF. (**a**) Toxicity screening without ATc 0.4 µM induction. (**b**) Toxicity screening with ATc induction. Error bars show ± standard deviation calculated using triplicate results. OD, optical density.

**Figure 6 antibiotics-12-01213-f006:**
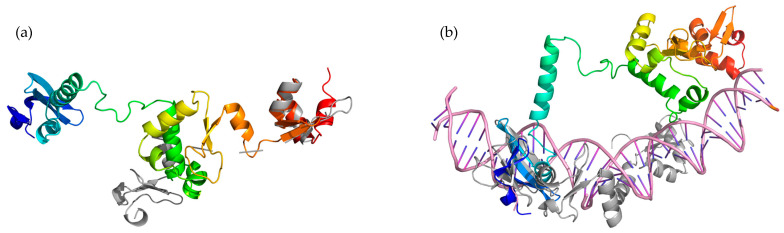
Structural and functional modelling of toxHPUFs Gp024 AlphaFold2 structural model (rainbow, N-terminus in blue, C-terminus in red) superimposed with *Bacillus* phage SPO1homing endonuclease I-HmuI (grey). (**a**) The model with the highest confidence from Phyre2. (**b**) The best model from HHpred, DNA structure is in pink.

**Figure 7 antibiotics-12-01213-f007:**
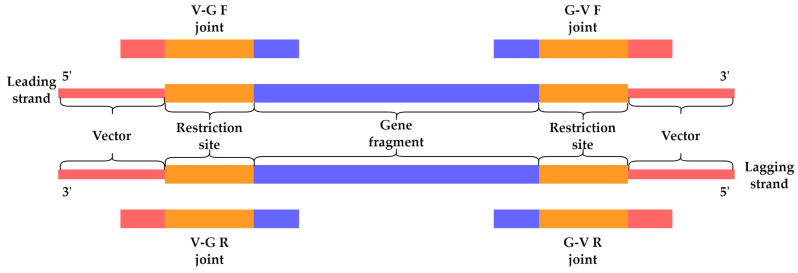
Illustration of the four ligation joint sequences used in the determination of sequence read coverage for each of the screened HPUFs (V, vector; G, gene fragment; F, forward; R, reverse). Adapted with permission from Nyhamar 2022 [12], University of Helsinki.

**Table 1 antibiotics-12-01213-t001:** Bacterial strains, phage and plasmids used in this study.

Name	Usage	Source
Stab21	Bacteriophage; the genome was used as a template for HPUF amplification	[14]
*E. coli* DH10B	HPUF cloning and NGS-based screening	Themo Fischer Scientific. GenBank accession no. NC_010473
*E. coli* DH5ɑ	HPUF cloning and screening with pBAD33	GenBank accession no. CP026085.1
*S. aureus* RN4220	HPUF cloning and screening with pRAB11N	GenBank accession no. CP076105.1
*S. aureus* Newman	ORF104 gene fragment isolation	GenBank accession no. NZ_CP087593.1
pCU1	Template for pCU1LK construction	[26] Kindly provided by Dr. Pentti Kuusela
pCU1LK	pCU1 containing multiple cloning sites between KpnI- PstI	This study
pBAD33	HPUF expression under arabinose-inducible promoter	[15]
pRMC2	Template for pRAB11 construction by insertion of second *tet* operator	[34]
pRAB11N	*E. coli/S. aureus* shuttle vector with tetracycline-inducible promoter for Stab21 HPUF expression	[27] Re-constructed in this study. GenBank accession no. JN635500

## Data Availability

Not applicable.

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
