# Peer review of "Discovery of Bactericidal Proteins from *Staphylococcus* Phage Stab21 Using a High-Throughput Screening Method"

_antibiotics, 2023, doi:10.3390/antibiotics12071213_

Round 1
Reviewer 1 Report
This study explores the potential toxicity and antimicrobial activity of selected gene products from the Staphylococcus phage Stab21 genome. The authors used various methods, including NGS-approach, drop tests, and growth curve analysis, to assess the toxic potential of 96 HPUFs (Highly Proliferated Uncharacterized Fragments) identified in Stab21. Furthermore, they investigated the inhibitory effect of these gene products on Escherichia coli and Staphylococcus aureus. Moreover, the paper predicts the functions and structural models of five selected gene products using various software tools.
Overall, this study presents valuable insights into the antimicrobial potential of selected gene products from the Staphylococcus phage Stab21 genome. The use of a diverse set of methodologies provides a robust assessment of toxicity and antimicrobial activity. Addressing the challenges of antibiotic resistance is of utmost importance, and this study contributes significantly to the field of antimicrobial agent development. The paper can be considered a valuable contribution to the scientific community.
Author Response
Comments and Suggestions for Authors
This study explores the potential toxicity and antimicrobial activity of selected gene products from the Staphylococcus phage Stab21 genome. The authors used various methods, including NGS-approach, drop tests, and growth curve analysis, to assess the toxic potential of 96 HPUFs (Highly Proliferated Uncharacterized Fragments) identified in Stab21. Furthermore, they investigated the inhibitory effect of these gene products on Escherichia coli and Staphylococcus aureus. Moreover, the paper predicts the functions and structural models of five selected gene products using various software tools.
Overall, this study presents valuable insights into the antimicrobial potential of selected gene products from the Staphylococcus phage Stab21 genome. The use of a diverse set of methodologies provides a robust assessment of toxicity and antimicrobial activity. Addressing the challenges of antibiotic resistance is of utmost importance, and this study contributes significantly to the field of antimicrobial agent development. The paper can be considered a valuable contribution to the scientific community.
Response: Thank you very much for your recognition of our work. Based on other reviewers’ and the academic editor’s comments, we added a few paragraphs to expend the background and to clarify the results. We also made a few corrections of the typos, and some additions of articles and conjunctions during the revision process.
Reviewer 2 Report
In this manuscript, this study focused on Hypothetical Proteins of Unknown Function (HPUFs) from a Staphylococcus phage Stab21. Author examined its HPUFs for bactericidal activity against E. coli using a Next Generation Sequencing (NGS)-based approach. Among the 96 HPUFs examined, five demonstrated cross-species toxicity towards E. coli, suggesting the presence of shared molecular targets between E. coli and S. aureus. It is clearly a big effort of the author for doing this study, this is already an improved study, the author did every possible experiment.
Author Response
Comments and Suggestions for Authors
In this manuscript, this study focused on Hypothetical Proteins of Unknown Function (HPUFs) from a Staphylococcus phage Stab21. Author examined its HPUFs for bactericidal activity against E. coli using a Next Generation Sequencing (NGS)-based approach. Among the 96 HPUFs examined, five demonstrated cross-species toxicity towards E. coli, suggesting the presence of shared molecular targets between E. coli and S. aureus. It is clearly a big effort of the author for doing this study, this is already an improved study, the author did every possible experiment.
Response: Thank you very much for your recognition of our work. Based on other reviewers’ and the academic editor’s comments, we added a few paragraphs to expend the background and to clarify the results. We also made a few corrections of the typos, and some additions of articles and conjunctions during the revision process.
Reviewer 3 Report
The current manuscript in-silico investigated the antimicrobial activities of several isolated proteins. The main significant point is trying these proteins on more diverse bacterial spp.
Language is Ok
Author Response
Comments and Suggestions for Authors
Point 1: (x) Moderate editing of English language required
Response 1: We now went through the whole manuscript to improve the English language. While responding to more concrete comments by the other reviewers and the academic editor, we added a few paragraphs to clarify the background information of this study and to explain the results more clearly. We tried our best to make them also meet a better English language standard. During the proof reading of the text, we corrected a few typos and added some articles and conjunctions as well.
Point 2: The current manuscript in-silico investigated the antimicrobial activities of several isolated proteins. The main significant point is trying these proteins on more diverse bacterial spp.
Response 2: In the lab, we have tested the toxHPUFs in both E. coli and S. aureus, which are very different species. Since the Stab21 phage has a wide host range against S. epidermidis, S. haemolyticus, S. xylosus and both methicillin resistant and sensitive S. aureus, it would be interesting to know whether the top toxHPUFs kills other Staphlococcus species. However, we are here to present a screening pipeline, as a proof-of-concept, which shows the streamlined NGS-based screening approach using E. coli as testing host can be used to screen various phage genomes regardless of phages’ hosts. Testing the toxHPUFs’ antimicrobial spectrum and functional study each toxHPUF would make a good follow-up story.
Reviewer 4 Report
The manuscript by Nyhamar et al characterizes five hypothetical proteins from phage Stab21 that are toxic. A NGS based approach was employed to screen for genes that can be toxic and the hits were further validated by spot assays. One of the genes gp024 was found to be similar to an endonuclease from a Bacillus phage. One interesting observation that warrants further investigation is the toxicity of anhydrotetracycline in S. aureus. Overall the work was well done and explained well.
Following are my comments that can improve the paper.
1. Instead of doing Phyre to establish homology the authors can do a BLASTP against the PDB database to establish homology.
2. There is no valuable information gained from comparing the structures predicted by different servers. Instead compare the predicted structure with the homolog.
Author Response
Comments and Suggestions for Authors
The manuscript by Nyhamar et al characterizes five hypothetical proteins from phage Stab21 that are toxic. A NGS based approach was employed to screen for genes that can be toxic and the hits were further validated by spot assays. One of the genes gp024 was found to be similar to an endonuclease from a Bacillus phage. One interesting observation that warrants further investigation is the toxicity of anhydrotetracycline in S. aureus. Overall the work was well done and explained well.
Following are my comments that can improve the paper.
Point 1. Instead of doing Phyre to establish homology the authors can do a BLASTP against the PDB database to establish homology.
Response 1: This is a very good point. The HPUFs in this study were hypothetical, as determined by blastp searches, which could not match the HPUF with an annotated, functional protein. For this reason, we had to apply other protein modelling tools to provide any clue on the potential functions of those HPUFs. In fact, we indeed performed BlastP searches of the top toxHPUFs against the PDB database, but no significant similarity was found. We suppose that as the searching algorithms among different servers are somewhat different, the searching results also vary though using the same database. Therefore, we think it would be worth to use a few different servers (e.g. Phyre2 and HHpred) to obtain more reliable predictions. We admit that there are other web-based tools such as i-TASSER, Swiss-Model, Robetta and Raptor etc. Phyre2 and HHpred have been routinely used in our lab, due to their applicability and performance. Therefore, we made the searches through these two webservers for this manuscript. As the manuscript is not focusing on the accessing the performances of different protein modelling tools, we did not use all existing tools.
Point 2. There is no valuable information gained from comparing the structures predicted by different servers. Instead compare the predicted structure with the homolog.
Response 2: We apologise for the confusions made in the manuscript, especially in Figure 6 legend. The two figures indeed showed the comparisons of the predicted structure of Gp024 with the homolog, from (a) the model obtained from Phyre2, and (b) obtained from HHpred.
To clarify this, we have now modified the figure 6 legend to: “Figure 6. Structural and functional modelling of toxHPUFs Gp024 AlphaFold2 structural model (rainbow, N-terminus in blue, C-terminus in red) superimposed with Bacillus phage SPO1homing endonuclease I-HmuI (grey). (a) The model with the highest confidence from Phyre2. (b) The best model from HHpred, DNA structure is in pink.”
Similarly, the supplementary figure S9 legend is now: “Figure S9. Superimposition of Gp081c (rainbow, N-terminus blue, C-terminus red) with either (a) the top Phyre2 model of Saccharomyces cerevisiae monopolin complex protein subunit CSM1 (grey) or (b) the best hit of HHPred functional models of Enterobacter phage 22 tail needle protein Gp26 (grey).”
We also revised the section 2.5. text to explain this result better:
“Eighty residues (37% sequence coverage) of Gp024 were modelled with 98.0% confidence to the C-terminus of Bacillus phage SPO1 homing endonuclease I-HmuI chain M (Protein Data Bank (PDB): c1u3eM). For g081c, a confidence of 85.5% was observed for 53 residues (48% sequence coverage), which matched to the Saccharomyces cerevisiae monopolin complex protein subunit CSM1...
The HHpred [11] software was also used to predict functions of the same toxHPUFs by searching each toxHPUF against PDB mmcif70.
The best hit for Gp024 was also Bacillus phage SPO1 homing endonuclease I-HmuI (PDB: 1U3E)…
The homologous models of Gp172 and Gp213 resulted in low probability and E-values.
When superimposing the truncated Bacillus phage SPO1 homing endonuclease I-HmuI (c1u3eM) obtained from the best Phyre2 model with the Gp024 AlphaFold2 model, the RMSD score was 1.581 Å between 138 atoms of the two proteins. Two C-terminal α-helixes both proteins were found overlapping with each other (Figure 6a). When superimposing the best homologous model of I-HmuI (PDB: 1U3E) from HHpred analyses with the predicted structure of Gp024, the RMSD score was 2.217 Å when aligning 181 atoms. The N-terminal DNA-binding surface of I-HmuI loosely overlapped with the N-terminus of Gp024 by two β-sheets and one α-helix (Figure 6b). Alignment of Gp081c with both Phyre2 and HHpred functional models resulted in RMSD scores over 8 Å (Figure S9).”